# Rewiring MAP kinases in *Saccharomyces cerevisiae* to regulate novel targets through ubiquitination

Benjamin Groves[1†], Arjun Khakhar[2†], Cory M Nadel[3], Richard G Gardner[3], Georg Seelig[1,4*]

[1]Department of Electrical Engineering, University of Washington, Seattle, United States; [2]Department of Bioengineering, University of Washington, Seattle, United States; [3]Department of Pharmacology, University of Washington, Seattle, United States; [4]Department of Computer Science and Engineering, University of Washington, Seattle, United States

**Abstract** Evolution has often copied and repurposed the mitogen-activated protein kinase (MAPK) signaling module. Understanding how connections form during evolution, in disease and across individuals requires knowledge of the basic tenets that govern kinase-substrate interactions. We identify criteria sufficient for establishing regulatory links between a MAPK and a non-native substrate. The yeast MAPK Fus3 and human MAPK ERK2 can be functionally redirected if only two conditions are met: the kinase and substrate contain matching interaction domains and the substrate includes a phospho-motif that can be phosphorylated by the kinase and recruit a downstream effector. We used a panel of interaction domains and phosphorylation-activated degradation motifs to demonstrate modular and scalable retargeting. We applied our approach to reshape the signaling behavior of an existing kinase pathway. Together, our results demonstrate that a MAPK can be largely defined by its interaction domains and compatible phospho-motifs and provide insight into how MAPK-substrate connections form.

*For correspondence: gseelig@uw.edu

†These authors contributed equally to this work

Competing interests: The authors declare that no competing interests exist.

## Introduction

The MAPK family of proteins is a ubiquitous signaling element in eukaryotes, and is essential to the function of a wide variety of cellular behaviors, from the regulation of differentiation and proliferation to stress responses and more (*Cargnello and Roux, 2011*); this diversity of functions has been made possible by the evolutionary expansion of the MAPK repertoire (*Caffrey et al., 1999*; *Manning et al., 2002*). For the expansion of the MAPK signaling module to have been feasible, it needed to be amenable to forming new kinase-substrate regulatory links, while at the same time having the capacity to avoid unwanted crosstalk. However, it still remains unclear what information is sufficient to create an entirely new set of regulatory interactions. One way to understand how potentially large numbers of novel regulatory links can be established is by developing a scalable method to create such links ourselves (*Elowitz and Lim, 2010*).

What are the core components necessary for the formation of a new – functional – kinase-substrate interaction? Following the association of the kinase and substrate, the amino acids in the immediate vicinity of the phosphorylated residue – together making up the 'phospho-motif' – help to dictate whether the substrate is phosphorylated by the kinase (*Mok et al., 2010*; *Howard et al., 2014*). However, it is the site that is phosphorylated – rather than the kinase itself – that mediates the functional outcome of kinase regulation. In particular, the phosphorylated phospho-motif can be recognized by a regulatory protein bearing a phospho-motif binding domain and control protein

**eLife digest** Nature has evolved a number of ways to link signals from a cell's environment, like the concentration of a hormone, to the behavior of that cell. These new connections often form by reusing certain common signaling components, such as mitogen-activated protein kinases. These enzymes – referred to as MAPKs for short – are activated by specific signals and alter the activity of target proteins in the cell by adding a phosphate group to them: a process called phosphorylation. These connections thus dictate how cells respond to their environments – and consequently, disruptions to the connections are a common source of disease.

Groves, Khakhar et al. set out to understand how connections can be made between a MAPK and a new target protein to gain insights into how these links emerge through evolution and how they might break in disease. Their approach focused on one of the ways that phosphorylation can alter the activity of a target protein: marking it for degradation. Experiments with budding yeast showed that a MAPK could only achieve this if two conditions are met. First, the target protein and kinase need to bind to each other. Second, the target needs to contain a site that when phosphorylated is subsequently recognized by the cell's protein degradation machinery.

By engineering proteins so that they fulfilled these two criteria, Groves, Khakhar et al. created new connections between a yeast MAPK called Fus3 or a human MAPK called ERK2 and a variety of targets. The results showed that the parts of the proteins involved in the interaction step could be completely separate from the parts that are involved in the phosphorylation step. This suggests that connections between kinases and their targets can be rewired simple by mixing together parts of other existing proteins. Finally, Groves, Khakhar et al. confirmed that engineered connections between kinases and targets could predictably change how yeast cells responded to a hormone that normally controls the yeast's reproductive cycle.

Together these results bring us one step closer to understanding how cells assemble the signaling pathways that they use to process information. However further work is needed to see if these findings can be generalized to other signaling components, and if so, to explore if new connections can be built to yield more complicated cellular behaviors.

localization or degradation among many other effects (*Seet et al., 2006*; *Bhattacharyya et al., 2006*).

Even before the kinase has a chance to interact with the phospho-motif, the two proteins must be colocalized (*Ubersax and Ferrell, 2007*). Residues apart from the kinase active site are frequently responsible for recognizing a substrate; indeed, several studies have sought to modify or replace these residues in a variety of kinases to redirect them to new – but still related – targets (*Skerker et al., 2008*; *Won et al., 2011*; *Grewal et al., 2006*). Adaptor proteins, such as synthetic scaffolds, have also been used to steer a kinase towards a particular native substrate (*Park et al., 2003*; *Whitaker et al., 2012*; *Hobert and Schepartz, 2012*; *Harris et al., 2001*). Regulation of a modified native substrate by a kinase can also be rescued using a pair of completely heterologous interaction domains (*Yadav et al., 2009*). These studies show that by controlling the colocalization of a kinase with a native – or closely related – substrate allows the functional regulation of that target. Taking it a step further, two groups have recently used native MAPK-interacting motifs – 'docking domains' – to allow several types of MAPKs in mammalian cells and yeast to regulate the nuclear localization of fluorescent reporters (*Regot et al., 2014*; *Durandau et al., 2015*). Although it is generally accepted that docking domains primarily control colocalization (*Sharrocks et al., 2000*), several studies suggest that binding may also serve to allosterically regulate the MAPK (*Chang et al., 2002*; *Heo et al., 2004*; *Zhou et al., 2006*; *Tokunaga et al., 2014*; *Bhattacharyya et al., 2006*). As such, the precise role of these interactions remains unclear. Regardless, the question of how completely new and orthogonal regulatory relationships are created remains.

Like the signaling modules that have been expanded in natural systems, engineered genetic circuits also rely on components that are amenable to rewiring. The creation of novel transcription factors has been successful in a large part because the necessary functional characteristics have been identified. Importantly, these characteristics can be embodied in distinct modular DNA and protein

domains, such as promoters, transcriptional-regulation domains, and DNA-binding domains – these domains can then be mixed and matched to yield the desired connectivity and regulation (*Khalil et al., 2012*; *Stanton et al., 2014*; *Kiani et al., 2014*; *Zalatan et al., 2015*; *Khakhar et al., 2015*). Although hurdles to creating large genetic circuits remain (*Brophy and Voigt, 2014*; *Cardinale and Arkin, 2012*), these parts have allowed scientists to construct and interrogate more complex engineered and naturally occurring genetic systems (*Prindle et al., 2012*; *Gilbert et al., 2014*). Unfortunately, our understanding of how to assemble modular post-translational signaling proteins lags behind. At the same time, recent work with engineered modular receptors expressed on T-cells has shown the considerable power of the ability to rationally design even relatively simple post-translational signaling systems (*Wu et al., 2015*; *Roybal et al., 2016*; *Morsut et al., 2016*).

Targeting a kinase to a new substrate is an essential step towards creating modular kinase signaling systems. As discussed above, Regot *et al.* and Durandau *et al.* have described an approach wherein a kinase-specific docking domain can be used to direct a particular kinase to a new substrate—a powerful tool for interrogating natural kinase signaling systems (*Regot et al., 2014*; *Durandau et al., 2015*). However, the number of naturally occurring kinase-substrate docking interactions inherently limits the scalability of the approach. For example, a given kinase 'module' cannot be reused in parallel signaling pathways, because it would not be able to distinguish between downstream targets in one pathway versus another. To overcome this limitation, it would be useful to be able to tease apart the 'targeting' module of the kinase from the 'enzymatic' module—and likewise, the 'targeting' and 'effector' modules of the substrate. If these functions can be defined as separable parts, the enzymatic module of a kinase would be available for reuse in orthogonal pathways, just by pairing it with unique targeting domains.

We have used simple, single-function modular protein domains to explicitly test the requirements for allowing a MAPK to regulate an arbitrary substrate protein. We utilized modular interaction domains to co-localize Fus3 – the terminal MAPK of the mating pathway of the yeast *Saccharomyces cerevisiae* – with a substrate of interest. To link phosphorylation of the substrate to a meaningful regulation event we utilized phosphorylation-activated ubiquitination-based signaling motifs—phosphodegrons. We re-targeted Fus3 to regulate several disparate proteins to determine the flexibility of the substrate design rules. Likewise, to determine whether this approach generalizes to other MAPKs, we retargeted a constitutively active version of the mammalian MAPK, ERK2, to functionally regulate a fluorescent reporter in yeast.

We explored the effect that synthetically implemented post-translational regulatory connections could have on the signaling of an endogenous kinase cascade in yeast. Our results demonstrate that these new connections can be used to alter the natural signaling behaviors, damping signal amplification and even yielding concentration-based band-pass filtering. Taken together, in this paper, we define a modular set of scalable components that can be utilized to rewire MAPKs to regulate proteins through ubiquitination. Attempting to rationally design new kinase-substrate regulatory links not only sheds light on the natural processes, but also serves as the foundation for the construction of synthetic kinase signaling pathways, and with them the control of cell behaviors in biomedical or biotechnological applications.

## Results

### Targeting a MAPK to phosphorylate and regulate a novel substrate

To test whether a direct interaction – along with a functional phospho-motif – can render an arbitrary protein a substrate for a MAPK, we used the yeast MAPK Fus3 to target and regulate a fluorescent reporter protein. Fus3 is easily triggered using the yeast mating pheromone, α-factor. α-factor signals to the central MAPK kinase cascade via a surface-associated receptor; signaling through the pathway activates Fus3, which in turn mediates signaling to a myriad of downstream effectors, directly regulating protein function and gene expression (*Figure 1A*) (*Bardwell, 2004*).

Given the important role ubiquitin-based degradation plays in signaling (*Hunter, 2007*; *Swaney et al., 2013*), we decided to use a phosphodegron as the regulated phospho-motif. Upon phosphorylation, the phosphodegron interacts with a specialized F-box protein – Cdc4 – to recruit the E3 ubiquitin ligase machinery (the SCF complex), which then marks the substrate for degradation by covalently attaching a poly-ubiquitin chain (*Figure 1A*) (*Skaar et al., 2013*). A phosphodegron

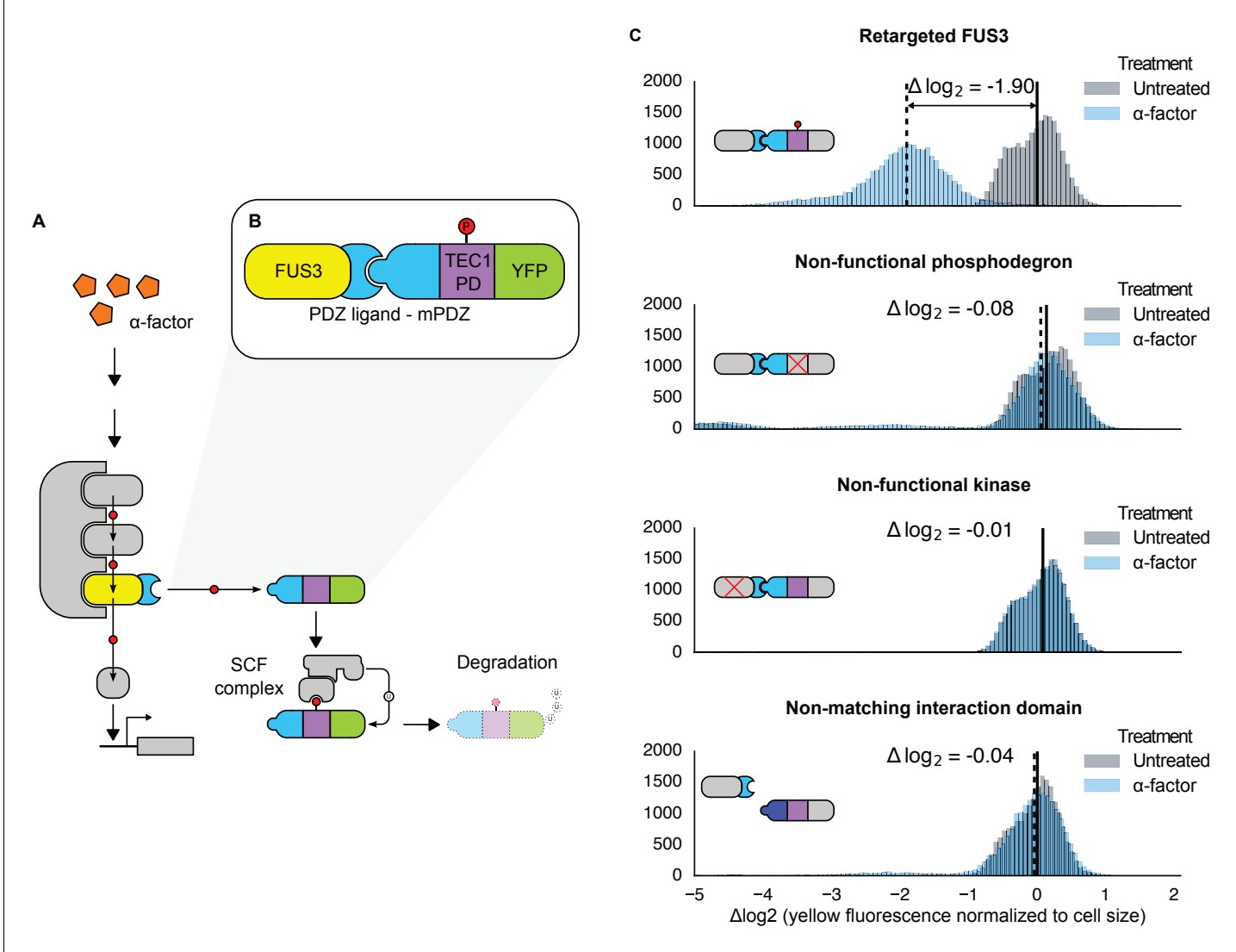

**Figure 1.** Rewiring the mating cascade MAPK, Fus3, to regulate the degradation of YFP. (**A**) The core components of the yeast mating cascade. The yeast mating factor – α-factor – triggers the sequential activation of the kinases Ste11 and Ste7 (rounded gray rectangles) followed by the MAPK, Fus3 (yellow). Arrows with red circles denote phosphorylation-mediated regulation. All three kinases are organized on the scaffold Ste5 (also gray). Among other effectors, Fus3 activates the transcription factor Ste12 (rounded gray box). (**B**) Fus3 targeted regulation of YFP (green). The colocalization was controlled by the addition of the mPDZ domain to YFP and a PDZ ligand to Fus3 (light blue). Degradation was mediated by the addition of a phosphodegron derived from the transcription factor Tec1 (purple). Upon activation of the mating pathway, Fus3 phosphorylates the phosphodegron fused to YFP, resulting in the recruitment of an E3 ubiquitin ligase and the ubiquitination and subsequent degradation of YFP. (**C**) Cells bearing the modified Fus3 and either the fully functional system, a reporter construct with an inactivated phosphodegron, a Fus3 with its kinase activity knocked out or an unmatched interaction domain (an SH3 domain instead of mPDZ) were grown to log phase and induced with 10 μM α-factor (blue histograms) or un-induced (gray histograms). Data shown are from 3 hrs post-induction. The vertical dashed black lines on the histograms represent medians of treated populations and solid black lines represent medians of untreated populations. In all figures, the fluorescence has been normalized to the cell size (see *Figure 1—figure supplement 1*). Full time-course experiments appear in the supplement to *Figure 2*.

The following figure supplements are available for figure 1:

**Figure supplement 1.** Reducing the variability of single-cell fluorescence by accounting for cell-to-cell variation in cell size.

**Figure supplement 2.** Western Analyses of degradation assays.

**Figure supplement 3.** Swapping interaction domains between kinase and substrate.

*Figure 1 continued*

**Figure supplement 4.** Fusing interaction domain to the native copy of the kinase.

has the added benefit of making a functional phosphorylation event easy to observe: if the substrate protein is a fluorescent reporter, such as YFP, phosphorylation and subsequent ubiquitination is followed by a decrease in YFP fluorescence. Thus, this approach is amenable to high-throughput measurements in a way that changes in localization may not be.

To start, we wanted to use a phosphodegron that was proven to be both functional and compatible with Fus3. The transcription factor Tec1 fulfills these criteria, as it has been shown to be both a substrate for Fus3 and Cdc4 (*Chou et al., 2004*; *Bao et al., 2010*)—thus, we chose a region of Tec1 that encompassed several residues up and downstream of the Cdc4 consensus sequence (37 residues, total) (*Nash et al., 2001*; *Orlicky et al., 2003*). Also, since Cdc4 primarily acts in the nucleus, we added a nuclear localization signal derived from SV40 large T-antigen to the N-terminus of the protein (*Blondel et al., 2000*; *Kalderon et al., 1984*). To complete our synthetic substrate, we needed to control its interaction with an engineered kinase. To this end, we added the mPDZ domain to the YFP-degron fusion, a modular protein interaction domain that has been used in a variety of different contexts (*Dueber et al., 2009*; *Moon et al., 2010*; *Ryu and Park, 2015*). To target Fus3 to the new substrate, we fused the complementary interaction domain, the PDZ ligand, to its C-terminus (*Figure 1B*). As in all the following experiments, these constructs were integrated as a single copy into the haploid yeast genome. Moreover, since we were only concerned with whether our modified Fus3 construct was able to functionally target our new YFP substrate – and not the behavior of other effectors downstream of the mating pathway – we did not remove the native *FUS3* gene—thus, our modified Fus3 construct operated in parallel with the native Fus3.

Following the induction of the mating pathway with 10 µM α-factor, we measured the YFP fluorescence of the cells using flow cytometry—to account for variation caused by cell-to-cell differences in cell size, we normalized the fluorescent signal by cell size (*Figure 1—figure supplement 1*). We observed a ~3.7-fold drop in the yeast strain containing both the Fus3-mPDZ ligand fusion and our new YFP-degron-mPDZ construct (*Figure 1C*). On the other hand, the drop in fluorescence was not observed when the phospho-acceptor residues in the degron (two threonine residues) were changed to methionine and alanine (*Bao et al., 2010*), when the catalytic site of the targeted kinase was inactivated with a K42R mutation (*Gartner et al., 1992*), or when the interaction domain fused to YFP was changed to an SH3 domain. The latter suggests that the Tec1 degron is not able to directly recruit Fus3 to the YFP construct on its own. Finally, we also found that the drop in YFP fusion protein level was sensitive to the presence of the proteasomal inhibitor MG132, strongly suggesting that the construct was indeed being tagged and actively degraded (*Figure 1—figure supplement 2*).

We also explored whether our rewiring approach was sensitive to which protein — the substrate or the kinase — the respective interaction domains were fused to. We built yeast strains in which the interaction domains were flipped—with the Fus3 kinase fused to the mPDZ domain and the YFP-degron fusion linked to a PDZ ligand. Following induction of the mating pathway with α-factor, we measured the YFP fluorescence of the cells using flow cytometry and observed qualitatively similar substrate degradation. However, the fold change observed three hours post induction for the swapped domains was approximately half that of the original orientation (*Figure 1—figure supplement 3*). This is likely due to the fusions affecting either protein expression or sterically interfering with the function of one of the involved enzymes. While these results demonstrate that this retargeting approach is largely modular, they also suggest that other characteristics of the fusions – such as how they affect translation or protein folding – may not be.

We also asked whether endogenous Fus3 could be re-targeted in the fashion described above. We found that by inserting the sequence encoding the PDZ ligand downstream of the native copy of the *FUS3* gene in the yeast genome, the native kinase could just as efficiently cause the degradation of the YFP substrate (*Figure 1—figure supplement 4*). These results – along with those discussed above – imply that an interaction domain and a phospho-motif are necessary and sufficient to target the regulation of a native signal transduction cascade to a substrate of choice.

## Expanding the repertoire of interaction domains

To determine how general our targeting approach is, we exchanged the mPDZ/PDZ ligand pair for unrelated pairs of modular protein interaction domains, both naturally derived and synthetic. We built variants of our Fus3-substrate pair with the naturally occurring SH3 domain or the synthetic SYNZIP domain (*Figure 2A*) (*Dueber et al., 2009*; *Thompson et al., 2012*). In both cases we observed significant reporter degradation, ~10.1-fold in the case of SH3 and ~4.7-fold for SYNZIP domains versus a control with a degron in which the two threonine residues in the Cdc4 binding site had been switched to a methionine or alanine (*Figures 2B* and *Figure 2—figure supplement 1*). These results confirm the flexible nature of the interactions that enable a productive kinase-substrate interaction.

We further tested our approach using a pair of inducible interaction domains derived from a plant hormone-sensing pathway. The association of the protein domains PYL and ABI can be controlled using the small-molecule plant hormone abscisic acid (ABA) (*Figure 2A*, right side) (*Liang et al., 2011*). When we fused these domains to Fus3 and our YFP-phosphodegron reporter, we observed a change in the fluorescent signal only when the concentration of ABA was 1 µM or higher (*Figure 2C*). These results provide additional evidence both that the kinase and substrate are indifferent as to the nature of their interaction, and that the targeting of the kinase to the substrate directly triggers the observed degradation, as the decrease in the YFP signal is correlated with ABA dose. However, it is important to note that the identity of the interaction domain fused to the YFP-phosphodegron target influenced the steady-state fluorescence of the reporter (*Figure 2—figure supplement 1*). Thus, even interaction domains with similar affinities may not have equivalent behaviors when used inside of cells.

We next investigated whether synthetic interaction domains enable multiple MAPKs to target independent substrates in parallel and in an orthogonal manner. We targeted one copy of Fus3 to an mCherry-phosphodegron reporter using a constitutive mPDZ-PDZ ligand interaction and a second copy of Fus3 to a YFP-phosphodegron reporter via the ABA inducible ABI-PYL interaction (*Figure 3A*). In the presence of α-factor alone only the mCherry signal was reduced, while the YFP value remained unchanged. Only when both α-factor and ABA were added, did we see a drop in the YFP signal (*Figure 3B and C*). From this perspective, the two Fus3 variants are analogous to orthologous MAPKs, with each targeting its own substrate.

However, we noticed that when there were two parallel MAPK-substrate systems in the same cell the net fold change of the ABA-sensitive YFP-phosphodegron reporter was moderately reduced compared to when it was present on its own—from ~2.15 to ~1.85 fold. We tested whether this decrease in efficiency was due to competition—either between the two copies of Fus3 for the pool of the activated upstream MAPK kinase, Ste7, or between the substrates for the ubiquitination/degradation machinery. To examine this question we constructed strains that expressed our standard system – one kinase targeting one substrate – and added either a competing copy of Fus3 or a competing substrate. In both experiments we observed a diminished response in YFP degradation in the presence of the competitor (*Figure 3—figure supplements 1* and *2*). Thus, it is likely that a confluence of factors – both saturation points as well as the less efficient ABA-induced interaction – contribute to the different levels of degradation observed for the mCherry and YFP reporters in this dual-targeting system.

The parallel synthetic kinases mimic the behaviors of a natural pair of yeast MAPKs, Fus3 and Kss1. Fus3 and Kss1 share many of the same targets, but also have distinct substrates, presumably as a result of the specialization of their preferences for related docking domains (*Reményi et al., 2005*). Likewise, the engineered system described above also retains the native targeting of Fus3, but uses distinct heterologous protein interaction domains to recognize unique targets.

## Exploration of alternative phosphodegrons

The ability to modulate the dynamics of MAPK-dependent degradation would be useful for reprogramming cell behaviors. We explored two strategies to modulate the degradation dynamics. First, we varied the number of phosphodegrons fused to the protein (*Figure 4A*). As we increased the number of phosphodegrons from one to three, we observed a concurrent increase in the rate of degradation of the reporter; adding more than three phosphodegrons to the reporter did not seem to affect the rate of degradation further (*Figure 4B*). In addition to changing the degradation

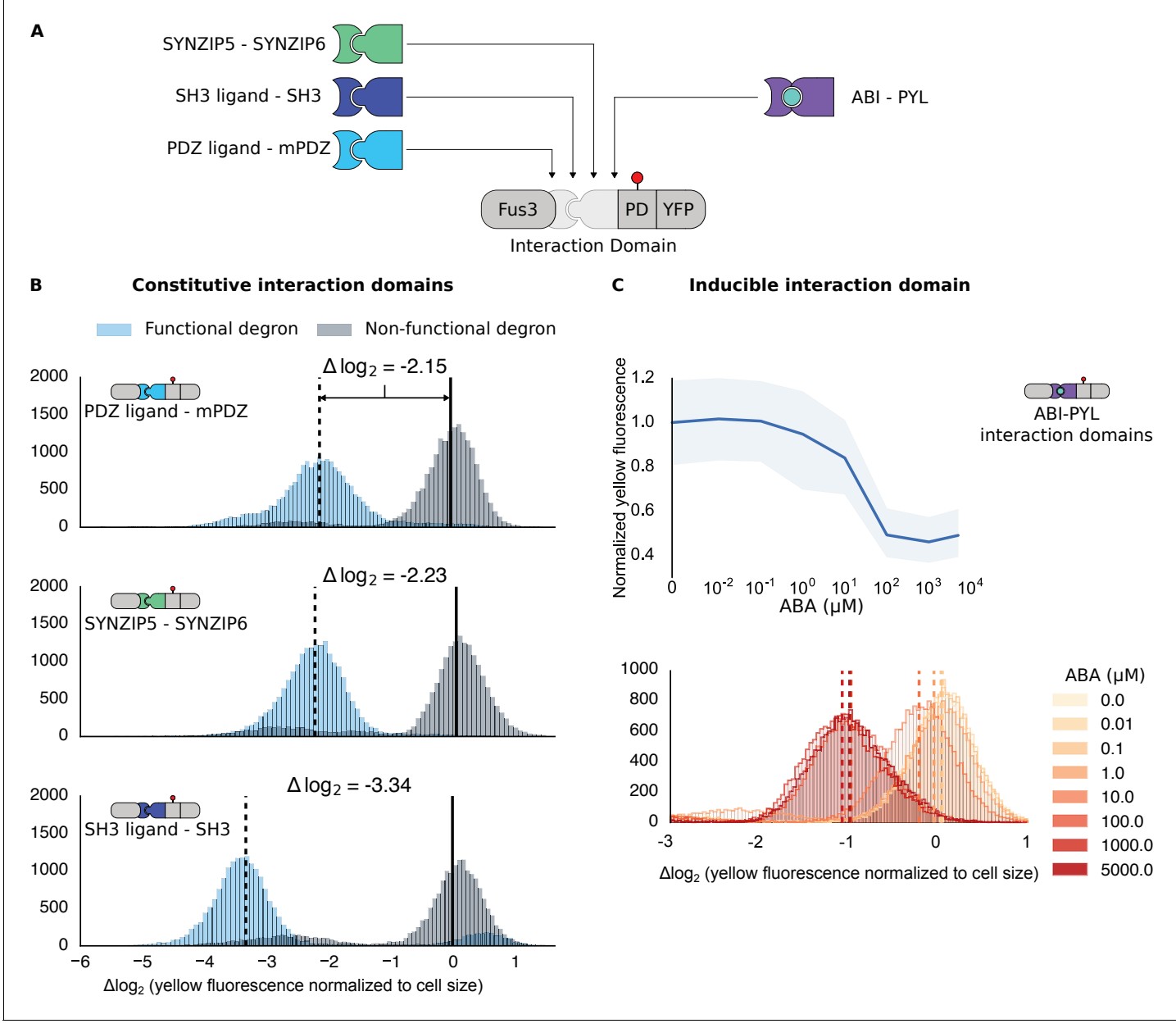

**Figure 2.** Demonstrating the flexibility and scalability of the system by varying interaction domains. (A) Variants of the different complementary interaction domains used. The constitutive interaction domains mPDZ, SH3 and SYNZIP are shown on the left; the ABA inducible ABI-PYL interaction domains appear on the right. (B) Comparison of YFP signal normalized by cell size from constructs bearing the indicated interaction domains along with either a functional (blue histograms) or non-functional (gray histograms) phosphodegron in yeast treated with 10 μM α-factor as in *Figure 1C*. The vertical dashed black lines on the histograms represent the medians of the populations with functional degrons whereas the solid black lines represent the median of the populations with non-functional degrons. (C) Median fluorescence – shaded regions cover the interquartile range – and population histograms of the YFP signal normalized to cell size from cells expressing the ABA inducible ABI-PYL interaction domains fused to Fus3 and YFP, respectively for a range of ABA concentrations. The raw time-course data corresponding to these endpoint observations can be found in *Figure 2— figure supplement 1*.

The following figure supplement is available for figure 2:

**Figure supplement 1.** Time course characterization of different interaction domain variants post induction with α-factor.

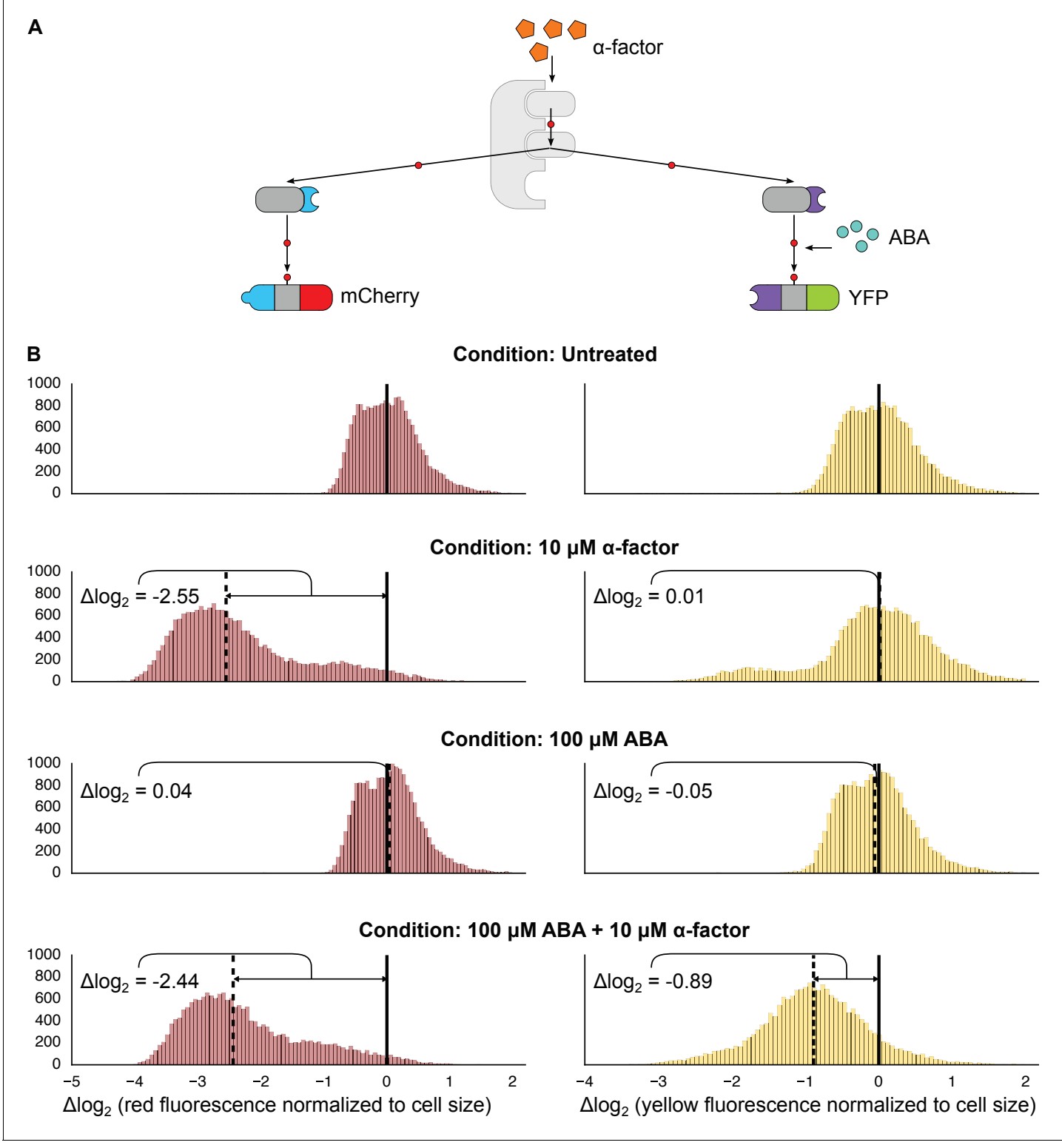

**Figure 3.** Targeting of orthogonal substrates. (**A**) Cells expressed two distinct forms of modified Fus3 and used either a constitutive interaction domain (left) or the ABA inducible domains (right) to target mCherry or YFP, respectively. (**B**) Population histograms of mCherry (left) and YFP (right) fluorescence normalized by cell size for cells under the indicated conditions—i.e. untreated, treated with 10 μM α-factor, treated with 100 μM ABA or both. The solid vertical black lines on the histograms represent the medians of the untreated populations and the dashed black lines represent the medians of the treated populations.

*Figure 3 continued on next page*

*Figure 3 continued*

The following figure supplements are available for figure 3:

**Figure supplement 1.** Competition between two Fus3 MAPKS with different interaction domains for MAPKK Ste7.

**Figure supplement 2.** Competition between mCherry and GFP when targeted by the same Fus3.

dynamics, increasing the number of phosphodegrons also decreased the steady state expression of

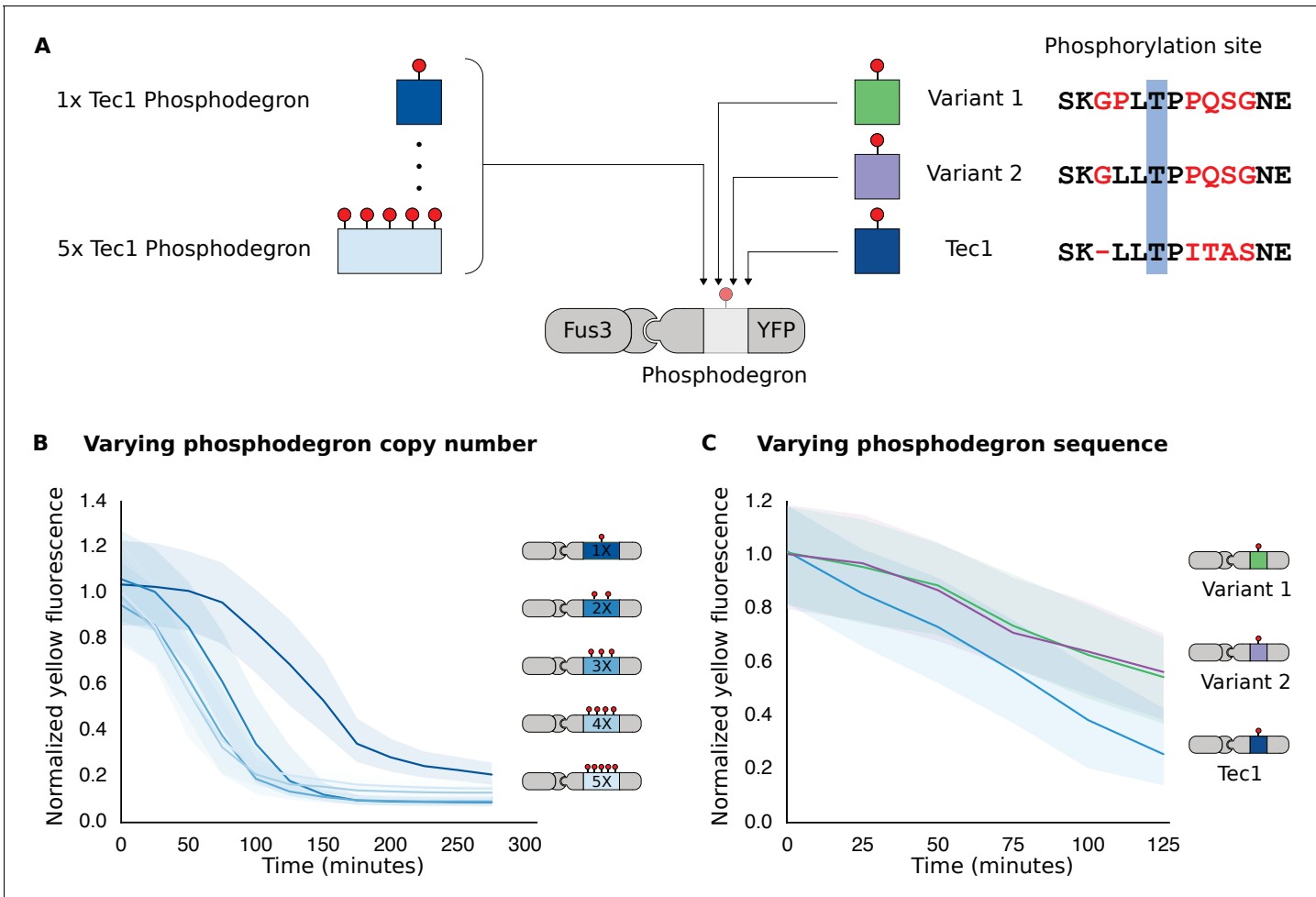

**Figure 4.** Modulating regulation by altering the number and sequence of phosphodegrons. (A) We varied either the phosphodegron number (left) or the sequence (right)—differing residues are red, the phosphorylated residue is highlighted in blue. (B) Time-course data of strains induced with 10 μM α-factor and expressing Fus3 targeting YFP reporters with one to five phosphodegrons. The fluorescence of each strain was normalized to cell size and then to its initial fluorescence. Data normalized only to cell size can be found in *Figure 3—figure supplement 1*. (C) Fus3 targeting of YFP substrates with the indicated phosphodegron sequence variants. As in B), the fluorescence of each strain is normalized to cell size and then against its initial fluorescence. Data normalized only to cell size can be found in *Figure 3—figure supplement 2*. The curves indicate the median values, while the shaded regions cover the interquartile range.

The following figure supplements are available for figure 4:

**Figure supplement 1.** Time course data of reporter variants with different numbers of phosphodegrons normalized by cell size.

**Figure supplement 2.** Time course data of reporter variants with different degron sequences normalized by cell size.

the reporter, possibly by multiplying the weaker interactions of the un-phosphorylated degron(s) with the degradation machinery (*Figure 4—figure supplement 1*).

We found that altering the amino acid sequence of the phosphodegron itself also changed the dynamics of degradation. We constructed two additional variants of the phosphodegron motif that more closely mimicked the amino acid sequence of the published 'consensus motif' for the WD40 domain of Cdc4 (*Figure 3A*) (*Nash et al., 2001*; *Orlicky et al., 2003*). The sequences of the two variants only differ at one site — two residues N-terminal of the phosphorylated threonine — where the Cdc4 consensus leucine was changed to a proline, an amino acid that is supposed to be preferred by Fus3 (*Mok et al., 2010*). Both variants had similar behaviors, with a similarly decreased rate of degradation relative to the phosphodegron derived from Tec1. (*Figure 4C*). These results suggest that phosphodegron design is flexible, and with more study it may become feasible to rationally tune their degradation dynamics. Moreover, the number of phosphodegrons is not limited to those found in nature. Taken together, these results demonstrate that our approach is applicable to several different phosphodegrons, and lays out a framework for how phosphodegrons may be used to alter degradation dynamics of a protein of interest.

## Retargeting the mammalian MAPK ERK2

We next swapped out the kinase module to test whether other MAPKs are also amenable to rewiring in the same manner. We focused on the human MAPK, ERK2—a widely studied kinase implicated in several pathologies, which has also been previously studied in the context of yeast (*O'Shaughnessy et al., 2011*). Native ERK2 has been shown to regulate protein stability via phosphodegrons; for example, a phosphodegron found in the protein MKP1 is targeted by ERK2 and subsequently tagged by the ubiquitin machinery and degraded (*Lin and Yang, 2006*). Our engineered substrate consisted of a 64 residue region surrounding the phosphodegron of MKP-1 fused to a YFP reporter. Rather than port the entire ERK2 signaling cascade into yeast, we used a constitutively active version of the MAPK created by fusing the upstream MAPK kinase – MEK1 – to ERK2 (*Robinson et al., 1998*). To enable the kinase-substrate interaction we fused the mPDZ domain and PDZ ligand to the substrate and MAPK, respectively (*Figure 5A*). We also included a construct missing the heterologous targeting domains to make sure that targeting was not simply due to direct interactions mediated by sequence elements surrounding the phosphodegron. Since the strains constitutively expressed both the engineered kinase and target, we measured the steady-state YFP fluorescence via flow cytometry. In strains with the active kinase targeting the functional YFP reporter, fluorescence did not rise above background levels (*Figure 5B*)—suggesting that the substrate is actively targeted, phosphorylated and then degraded. Fluorescence was significantly higher in control strains where the interaction domain, the phosphodegron or both were missing or inactivated (*Figure 5B*). These results indicate that the interaction domains and the phosphodegron are necessary and sufficient for retargeting the regulation of ERK2. Importantly, these results also strongly suggest that this rewiring approach is potentially applicable to a wide range of MAPKs.

## Modifying MAPK cascade signal processing

Thus far, we have described the retargeting of MAPKs to synthetic targets such as fluorescent proteins, which double as the readout for kinase activity. Next, we asked whether MAPKs could be targeted to arbitrary endogenous substrates and – more specifically – whether this approach can be used to modify the response of an existing signaling pathway. To answer these questions, we targeted Fus3 to up- and downstream elements in the yeast mating cascade, including the kinase Ste7, the scaffold protein Ste5, and the transcription factor Ste12. We built a total of six yeast strains containing the synthetic kinase-substrate pairs. Three of these strains constitutively expressed Fus3 with a PDZ interaction domain, while the other three expressed a version of Fus3 with a non-matching interaction domain. All of the strains included one of the mating cascade proteins – Ste5, Ste7 or Ste12 – fused to a complementary interaction domain and the Tec1 phosphodegron. The interaction domain and phosphodegron were inserted into the native genomic locus of the protein of interest. The Fus1 gene, whose expression is activated by the mating pathway upon induction with α-factor, was fused to YFP to provide an independent readout for pathway activation.

We chose these specific target proteins because their regulation by Fus3 results in interesting regulatory topologies. Specifically, Fus3-mediated degradation of Ste7 and Ste5 are examples of

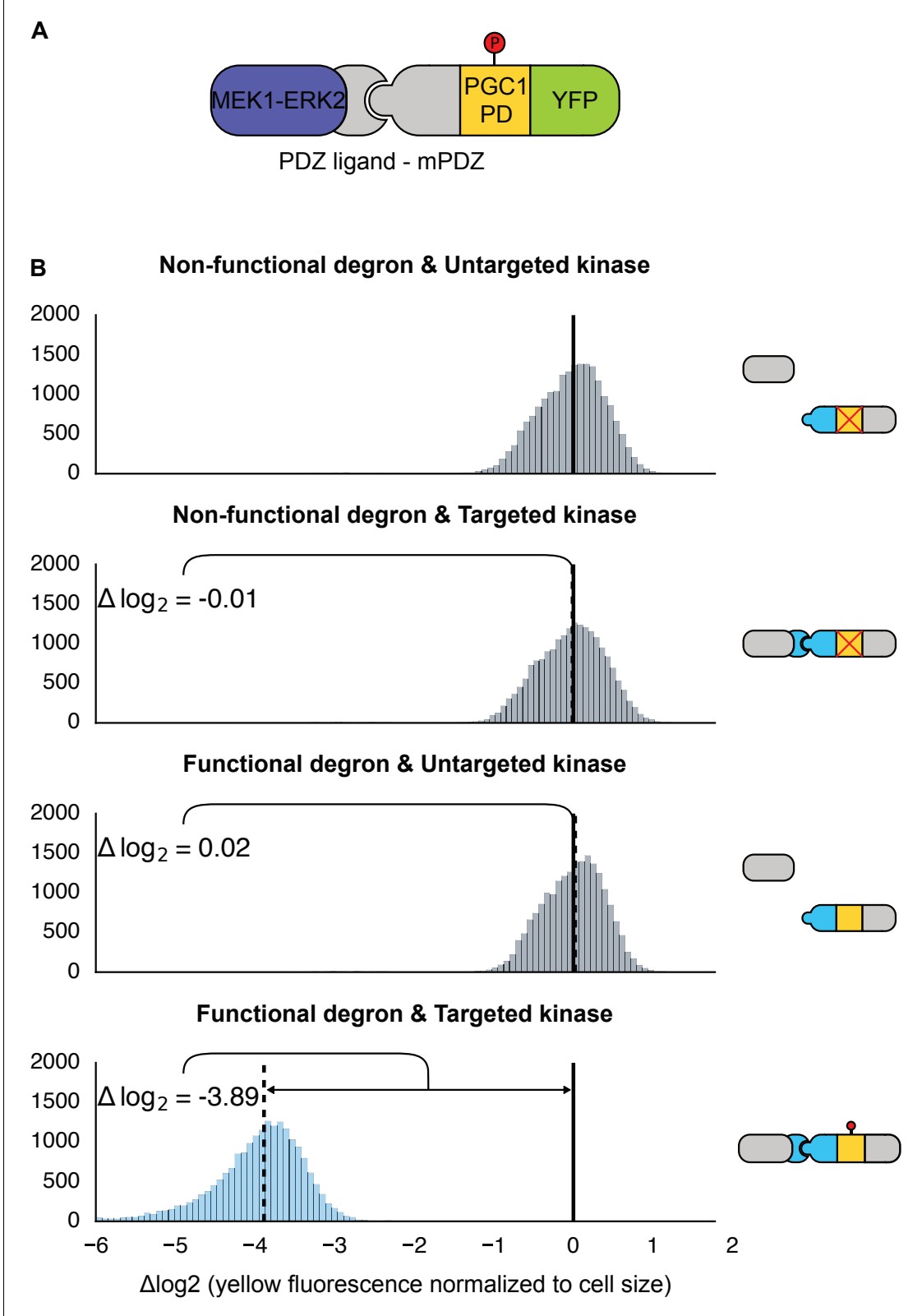

**Figure 5.** Retargeting the mammalian MAPK, ERK2. (**A**) As with Fus3, the human MAPK, ERK2, was targeted to a YFP reporter (green) via an interaction between the mPDZ domain and the PDZ ligand. A phosphodegron (yellow) fused to the YFP reporter was derived from the mammalian MKP-1. ERK2 was rendered constitutively active by fusing it to a constitutively active form of MEK1 (purple). (**B**) Population histograms of YFP fluorescence normalized by cell size of yeast strains in log phase growth with active ERK2 targeted to YFP with a functional phosphodegron (blue histogram). Controls strains

*Figure 5 continued on next page*

*Figure 5 continued*

with an inactive phosphodegron fused to YFP and/or an untargeted version of the kinase were also tested (gray histograms). The solid vertical black lines on the histograms represent the medians of the first histogram – the untargeted kinase paired with the non-functional degron – and the dashed black lines represent the medians of each subsequent population.

negative feedback loops, while the degradation of Ste12 results in an incoherent feed-forward loop (*Figure 6A–C*). Such regulatory links can be used to fundamentally alter the signal processing properties of native pathways (*Alon, 2007*; *Bashor et al., 2008*). Of note, this is the first time that purely post-translational feedback loops have been used to re-engineer signaling.

To determine the impact of negative feedback, we measured the fluorescence output of the pathway following induction with varying levels of α-factor. Relative to the untargeted controls, the negative feedback through Ste5 or Ste7 reduced the maximal pathway activation in those backgrounds by ~60% and ~45%, respectively. The apparent Hill coefficients ($n_H$) were also moderately changed with negative feedback compared to the untargeted kinase controls—when Ste5 was the target, $n_H$ increased from 1.5 to 1.9, while when Ste7 was negatively regulated $n_H$ remained 1.6 (*Figure 6A and B*). These values are qualitatively consistent with but slightly higher than the sensitivities reported previously for a system with negative feedback realized through transcription and recruitment of a phosphatase in the yeast mating cascade (*Bashor et al., 2008*).

The increase in pathway sensitivity observed for negative feedback applied to Ste5 is surprising (*Kholodenko, 2000*). However, the response of a scaffolded signaling cascade is highly sensitive to the concentration of the scaffold protein—with a reduction of the scaffold concentration resulting in an increase in the sensitivity of the cascade (*Levchenko et al., 2000*). Although a more detailed analysis is required, this observation suggests a potential explanation for the observed increase in the apparent Hill coefficient. However, we also note different fusion proteins are required for each experiment and that these protein modifications alone can result in changes of the pathway sensitivity—e.g. by changing the concentrations of pathway components (*O'Shaughnessy et al., 2011*). For example, the non-feedback controls in the three experiments shown in *Figure 6* have apparent Hill coefficients of $n_H$ =1.5, 1.6 and 2.1.

In the incoherent feed-forward loop – created by having Fus3 both activate and inhibit the transcription factor Ste12 – we find that the inhibitory connection dominates at all levels of induction resulting in a complete elimination of the downstream response (*Figure 6C* and *Figure 6—figure supplement 1*). However, as we will show next, more interesting behaviors are possible in a slightly more complex incoherent feed-forward loop.

Hybrid regulatory schemes that occur at the level of both transcription and translation are often observed in nature and further enrich the available behaviors in the design of engineered biological circuits (*Yeger-Lotem et al., 2004*; *Mishra et al., 2014*). By putting the YFP-phosphodegron-mPDZ domain fusion protein under the control of the mating pathway-controllable promoter – pFUS1 – we created a simple incoherent feed-forward circuit regulated at the level of both transcription and translation. Such a 'type 3' incoherent feed-forward loop design can produce pulses and other behaviors, depending on the design parameters (*Mangan and Alon, 2003*). A phenomenological model of a hybrid incoherent feed-forward loop is included in Appendix 1. We performed time-course experiments over a range of α-factor concentrations (*Figure 7B* and *Figure 7—figure supplement 1*). In cells containing the feed-forward loop the fluorescent signal initially increased sharply as the α-factor concentration was increased from 0.1 μM to ~1 μM—however, induction with concentrations of α-factor higher than 1 μM resulted in decreasing levels of YFP fluorescence. The incoherent feed-forward loop thus created a concentration-based band-pass filter for the α-factor input. In a control where the phosphodegron fused to YFP was broken we observe the normal signal amplification behavior of the mating cascade (*Figure 7A*)—thus, it is the targeted regulation of YFP by Fus3 that caused the band-pass-like behavior.

Taken together, these results demonstrate that re-targeted kinases can be used to modulate the behavior of signaling cascades through variety of circuit designs, including negative feedback and incoherent feed-forward loops. The data also highlight the utility of using this rewiring approach to study the effects of kinase-directed ubiquitination-based regulation, which occur extensively in nature (*Swaney et al., 2013*; *Beltrao et al., 2012*) and adds to the available tools for the study of

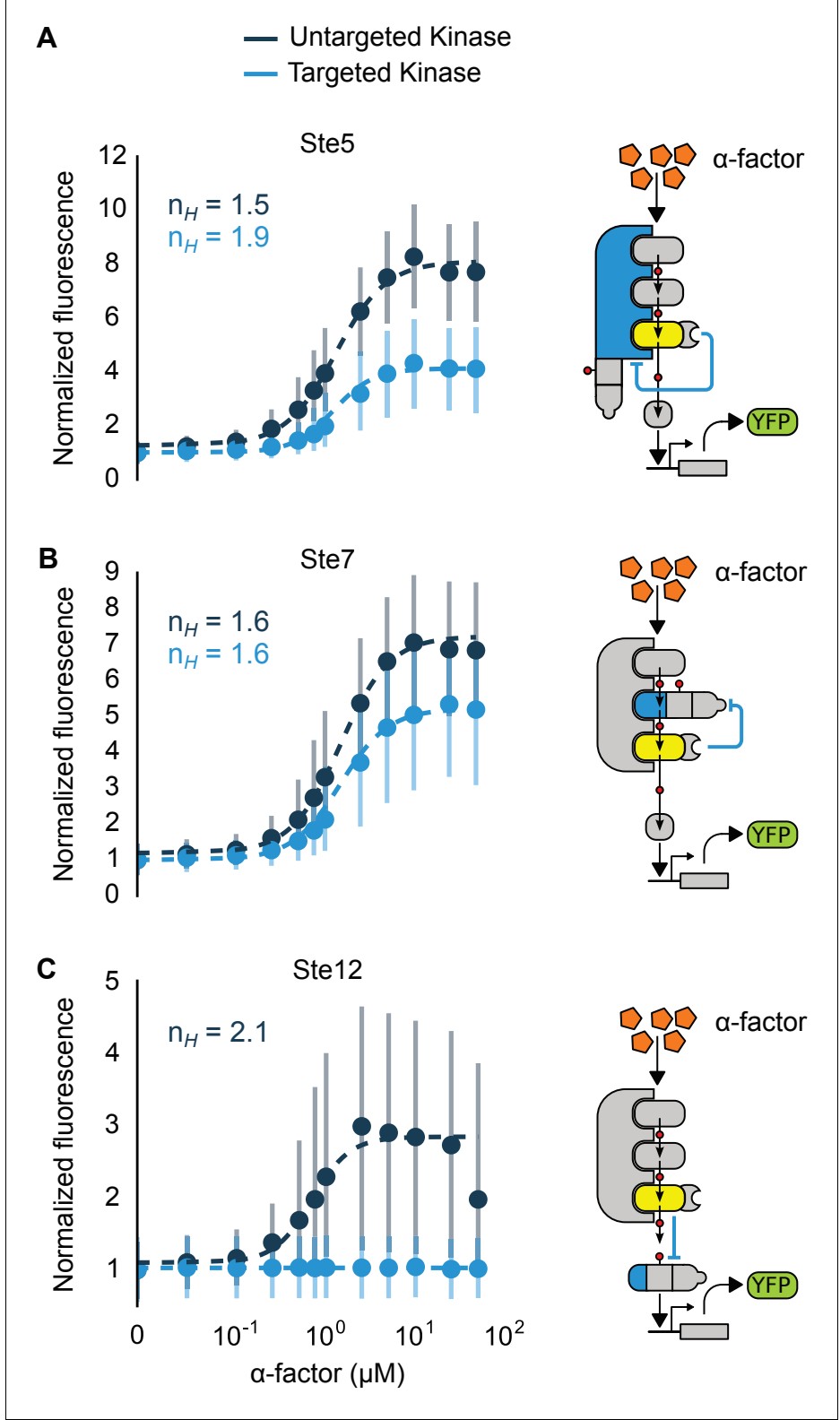

**Figure 6.** Implementation of negative feedback and feed-forward signaling topologies using a rewired MAPK. (A–C) Plots and schematics that depict the relationship between the α-factor input and the YFP reporter for yeast strains with synthetic post-translational negative feedback or feed-forward loops. Fus3 (yellow) was rewired to target (A) the scaffold Ste5, (B) the kinase Ste7 or (C) the transcription factor Ste12 (all depicted in light blue)–in each case, the endogenous copies of these proteins were modified by inserting a phosphodegron and a

*Figure 6 continued on next page*

*Figure 6 continued*

complementary interaction domain at their C-terminus. Plots of the median fluorescence of the YFP reporter – under the control of the mating-specific pFUS1 promoter – normalized to cell size for increasing concentrations of α-factor. Data from control strains with an untargeted kinase – and thus no feedback/feed-forward control – are shown in dark blue. Points indicate the median values at each α-factor concentration, while the vertical bars cover the interquartile range of the data. The data from both the no feedback and feedback conditions were used to determine the parameter values used with the formula: $A + B\frac{[\alpha]^n}{1+C[\alpha]^n}$ – where $C$ was fixed between the two data sets. $n$ and $[\alpha]$ are the hill coefficient and the α-factor concentration, respectively. Fits are plotted as dashed lines. Time courses of the same strains treated with 10 µM α-factor are shown in *Figure 6—figure supplement 1*.

The following figure supplement is available for figure 6:

**Figure supplement 1.** Time course characterization of Negative feedback topologies.

this pervasive mode of signaling (*Schneekloth et al., 2004*; *Melchionna and Cattaneo, 2007*; *Bonger et al., 2011*; *Neklesa et al., 2011*).

## Discussion

Here we have demonstrated that MAPK-directed ubiquitin-based signaling can be rewired to regulate a protein of choice. The addition of sets of two modular components is sufficient to rewire a MAPK to regulate any protein of interest—a complementary set of protein interaction domains and a phosphodegron. Natively, MAPKs are co-localized with their substrates via an interaction between the docking peptide of a substrate and a set of residues on the surface of the MAPK; it has been hypothesized that this interaction may be necessary to catalytically unlock the kinase (*Chang et al.,*

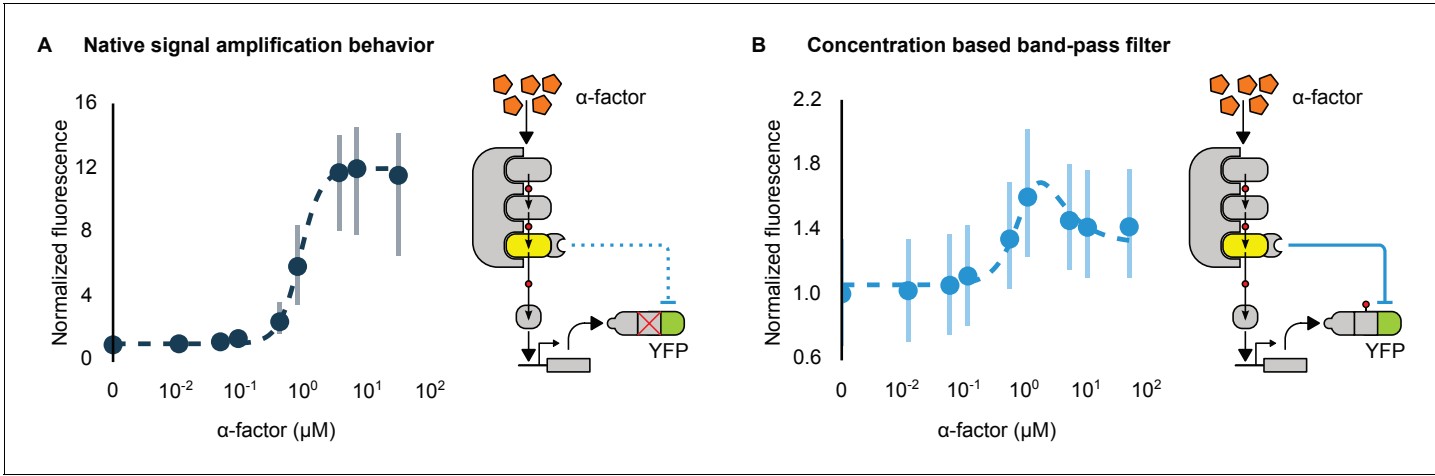

**Figure 7.** Conversion of a native amplifier to a band-pass filter. (A, B) The relationship between the α-factor input and YFP expression – driven by the mating pathway-specific promoter pFUS1 – for strains without and with a synthetic post-translational incoherent feed-forward loop. Induction of the mating pathway activated a modified Fus3 (yellow) that indirectly up-regulates the expression of a YFP reporter (green) fused to a phosphodegron. An interaction between the Fus3 and the YFP-degron reporter was enabled via complementary interaction domains. In one case (A) the phosphodegron was mutated and inactive, while in the other (B) it was fully functional. The points indicate the median YFP signal – normalized by cell size and then to the untreated condition – in yeast strains in log phase growth treated with the indicated concentration of α-factor. The error bars depict the interquartile range of the population data. Dashed lines are fits to the equation $A + B\frac{[\alpha]^n}{1+C[\alpha]^n}\frac{1+E[\alpha]}{1+D[\alpha]}$ – model derivation and fitting are described in more detail in Appendix 1. Time-course data is shown in *Figure 7—figure supplement 1*.

The following figure supplement is available for figure 7:

**Figure supplement 1.** Time course of dose response behavior to α-factor induction of yeast strains with (top row) and without (bottom row) the mating cascade modified with an incoherent feed forward loop.

*2002*; *Heo et al., 2004*; *Zhou et al., 2006*; *Tokunaga et al., 2014*). Our results suggest that while these domains may have some allosteric properties, simply co-localizing an active MAPK with a protein bearing a compatible amino acid motif that can be phosphorylated is sufficient for the functional regulation of the protein.

One implication of our results is that the evolution of new connections within MAPK regulation networks is only constrained by the two criteria discussed above—namely the appearance of 1) an accessible phospho-motif; and, 2) a protein-protein interaction strong enough to co-localize the new kinase-substrate pair. The proteomes of *Saccharomyces cerevisiae* and humans have ~3500 and >50,000 phosphorylation sites, respectively (*Hornbeck et al., 2015*, *2012*; *Beltrao et al., 2009*). The amino acid composition of the surrounding phospho-motifs is constrained by the residues in and around the kinase active site; as such their length is generally fairly short—on the order of four amino acids on either side of the phosphorylated residue (*Ubersax and Ferrell, 2007*). With such a short length, and given the degeneracy of the recognition requirements (*Mok et al., 2010*), the probability that new phospho-motifs will appear by chance is high. Indeed, many human SNPs – both those associated with disease as well as apparently healthy individual variation – have been observed to create and destroy verified phospho-motifs (*Hornbeck et al., 2015*; *Reimand and Bader, 2013*; *Ryu et al., 2009*).

Many protein interactions occur between short, linear stretches of amino acids and protein domains, the classic examples being the PDZ and SH3 domains, but the binding of docking domains to the surfaces of MAPKs also belongs to this class (*Harris and Lim, 2001*; *Mayer, 2001*; *Reményi et al., 2006*; *Van Roey et al., 2014*). Like phospho-motifs, these short motifs can appear spontaneously during evolution (*Neduva and Linear motifs, 2005*; *Davey et al., 2015*; *Beltrao and Serrano, 2007*). Given that both phospho-motifs and short, linear interaction peptides are degenerate, common and short it is interesting to consider what constrains the formation of a new, functional connection between a kinase and a substrate—i.e. whether it is the formation of phospho-motifs or of protein-protein interactions that is rate limiting. This may be addressed by future studies.

Our creation of modular components for kinase signaling may help recapitulate the success modular transcriptional circuits have enjoyed (*Stanton et al., 2014*; *Kiani et al., 2014*; *Zalatan et al., 2015*; *Prindle et al., 2012*). However, while our approach is a powerful tool it does have certain limitations. For instance, our system requires that a phosphodegron be known, and its cognate F-box be expressed for ubiquitination to occur. We demonstrate one way in which this problem may be addressed, i.e. by the design of new phosphodegrons based on the consensus sequences of the MAPK and F-box. Another consideration in any protein-engineering endeavor is the effect that various protein fusions have on expression—indeed we noted in our experiments that fusion of additional domains to MAPKs or their substrates altered the expression levels. These altered expression levels affect the behavior of kinase cascades, and so a sufficiently diverse set of modules need to be defined and characterized to make the desired behavior(s) achievable. Thus, the scalability afforded by the use of modular interaction domains comes at the potential price of altered protein expression. In contrast, using docking domains for co-localization obviates engineering the kinase, but is not a scalable rewiring approach. Finally, more work is necessary to render other kinase families 'engineerable'. Still the flexibility and scalability of kinase-substrate interactions demonstrated through our work lays a comprehensive foundation for future attempts to understand and re-engineer the signaling behavior of cells.

## Materials and methods

### Strain construction

All strains were built using a W303a background into which each synthetic construct was integrated at either the URA, HIS, TRP or LEU genomic loci. The plasmids used to generate the strains are listed in *Supplementary file 1*. The YFP reporter constructs were built by fusing an SV40 nuclear localization tag, an interaction domain and a phosphodegron in tandem to the YFP protein separated by 12 amino acid long glycine-serine linkers. The retargeted kinase constructs were built by fusing a complementary interaction domain to the kinase, also separated by a 12 amino acid glycine-serine linker. The strong constitutive promoter derived from the native TDH3 gene was used to drive expression

of the constructs. For all examples of the system that involved the yeast mating cascade, the kinase used was the MAPK *FUS3*. For the system that demonstrated mammalian MAPK retargeting, the kinase utilized was a constitutively active version of MEK1 fused to ERK2 and an interaction domain. For the feedback and feed-forward strains YFP was fused in tandem with the FUS1 gene, whose expression was activated by the mating pathway, to act as a reporter. These strains also had a copy of *FUS3* fused to an interaction domain integrated into the genome. In the case of the negative feedback and the feed-forward strains the genomic copies of Ste5, Ste7 and Ste12 were fused to an interaction domain, a phosphodegron and an mCherry reporter. The incoherent feed-forward strains were identical except that the expression of the YFP-nuclear localization tag-phosphodegron-interaction domain fusion was driven from a FUS1 promoter.

## Cytometry

All cytometry measurements in experiments just measuring YFP expression were acquired with an Accuri C6 cytometer with attached CSampler apparatus using 488 nm and 640 nm excitation lasers and a 533 nm (FL-1: YFP/GFP) emission filter (BD Biosciences). In those experiments that included mCherry, we used a MACSQuant VYB (Miltenyi Biotec), with 405, 488 and 561 nm excitation lasers and 561 nm (FSC), 525 nm (YFP) and 615 nm (mCherry) emission filters. Synthetic complete growth medium was used in all experiments. Experiments involving time course data were taken during log phase via the following preparation: 16 hrs of overnight growth in the synthetic complete medium in a 30°C shaker incubator followed by 1:100 dilution into fresh, room-temperature medium. After 5 hrs of growth at 30°C, 100 µL aliquots were read periodically – with 10 thousand events collected for every read – until the completion of the experiment. In all cases where Fus3 was being retargeted, the yeast cultures were induced with α-factor 5 hrs post-dilution. For experiments involving dose response behavior, cultures were grown overnight, then diluted down in the morning 1:100 in fresh media and grown for 5 hrs to log phase. They were then induced with α-factor, as well as other inducers like ABA in some cases, and allowed to grow for between two to six hours depending on the experiment and then read on the cytometer. As the MEK-ERK2 fusion is constitutively active no inducer was necessary (*Robinson et al., 1998*).

## Cytometry data analysis

Data were analyzed using custom python scripts and FCSParser and Seaborn libraries (DOI: 10.5281/zenodo.45133) using the following steps: (1) Anomalies – such as bubbles – were identified by plotting and visually inspecting the FSC-A value versus the time each cell was collected for each well. (2) To prevent the creation of NA values when the data was log transformed any 0 values in the data collected from the Accuri C6 cytometer were converted to 1. Since data collected on the MACSQuant VYB can fall below 0, all the data was normalized by adding the absolute value of the lowest value collected that day to the raw values and then adding 1. (3) To control for the effects of cells size, the fluorescence values for each event were then divided by the FSC-A value for that event. All reported data is the aggregate of at least two technical replicates performed on consecutive days. The fits presented in *Figures 6* and *7* were performed using custom python scripts.

## Degradation assays

10 mL cultures of yeast strains expressing untargeted control substrates or targeted test substrates were grown at 30°C in YEPD medium to approximately $1*10^7$ cells/mL. Cells were incubated with DMSO or the proteasome inhibitor MG132 (25 µg/mL) for 30 min prior to addition of α-factor or vehicle control for an additional 10 min. Cycloheximide was then added to a concentration of 50 µg/mL and cells were harvested by centrifugation at the denoted time points. Pellets were lysed in 200 µL SUMEB buffer (8 M urea, 10 mM MOPS, 10 mM EDTA, 1% SDS, 0.01% bromo- phenol blue, pH 6.8) by vortexing with acid washed beads for 5 min at 25°C. Lysate was clarified by centrifugation at 13000 rpm for 5 min and subjected to western analysis.

## Western analyses

Protein lysates were resolved by SDS-PAGE using 4–20% gradient gels (Lonza). Western analyses were performed with rabbit anti-GFP (1:2500) or mouse anti-ubiquitin antiserum (1:10).

## Acknowledgements

We are grateful to Clare Campbell, Michelle Parks, and Klavins lab technicians for technical support and to Alberto Carignano for help with modeling. This work was supported by the National Science Foundation (NSF) Awards EFMA 1137266 and CCF 1317653 to GS and an Institute for Protein Design Washington Research Foundation Innovation Fellowship to BG.

## Additional information

### Funding

| Funder | Grant reference number | Author |
|---|---|---|
| National Science Foundation | EFMA-1137266 | Georg Seelig |
| WRF-IPD Innovations Fellows Program | | Benjamin Groves |
| National Science Foundation | CCF-1317653 | Georg Seelig |

The funders had no role in study design, data collection and interpretation, or the decision to submit the work for publication.

### Author contributions

BG, AK, Conception and design, Acquisition of data, Analysis and interpretation of data, Drafting or revising the article; CMN, Designed and performed biochemical degradation assay; RGG, Designed biochemical degradation assay; GS, Conception and design, Analysis and interpretation of data, Drafting or revising the article

### Author ORCIDs

Benjamin Groves, http://orcid.org/0000-0002-4827-5500
Arjun Khakhar, http://orcid.org/0000-0002-4676-6533
Georg Seelig, http://orcid.org/0000-0002-3163-8782

## Additional files

### Supplementary files
• Supplementary file 1. Table of yeast strains.

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

## Appendix 1

# An analytical model for the incoherent feed-forward lLoop

**Mangan and Alon (2003)** initially classified the different types of feedforward loop architectures and pointed out that Type 3 IFFLs can robustly generate pulses in gene expression. Here, we present a simple analytical model of the specific type 3 IFFL introduced in the main text, following an approach outlined by **Shvartsman and Baker (2012)**. The purpose of this model is to provide an intuition for the origin of the bandpass behavior that we observed in our experiments, rather than to give a detailed mechanistic study of the actual signaling cascade. We only consider three species in our model, namely Fus3, Ste12 and GFP. Fus3 is activated (indirectly) by $\alpha$-factor and phosphorylates Ste12. Phosphorylated Ste12 acts as a transcriptional activator for GFP. Through a synthetic interaction Fus3 also directly phosphorylates the Tec1 phosphodegron fused to GFP, triggering the degradation of GFP.

For both Ste12 and Fus3 we only explicitly model the phosphorylated, active form. We use saturating functions to describe the phosphorylation and hence activation of Fus3 by $\alpha$-factor and of Ste12 by Fus3. A similar function is used to model the production of GFP in response to Ste12. Degradation of GFP can be due to the interaction with Fus3 or due to other causes and the two processes are modeled independently. The full model is given by:

$$\frac{d}{dt}[Fus3] = \beta_3 \frac{[\alpha]}{K_\alpha + [\alpha]} - \gamma_3[Fus3], \tag{1}$$

$$\frac{d}{dt}[Ste12] = \beta_{12} \frac{[Fus3]}{K_3 + [Fus3]} - \gamma_{12}[Ste12], \tag{2}$$

$$\frac{d}{dt}[GFP] = \beta_{GFP} \frac{[Ste12]}{K_{12} + [Ste12]} - \gamma_{GFP}[GFP] - \Gamma[Fus3][GFP]. \tag{3}$$

Here, $\beta_3$, $\beta_{12}$ and $\beta_{GFP}$ are the maximal production rates while, $\gamma_3$, $\gamma_{12}$ and $\gamma_{GFP}$ are degradation rates. The paramter $\Gamma$ measures the strength of the Fus3-mediated degradation. If we assume that the system is in steady state we obtain three coupled equations for the three variables:

$$[Fus3]_{ss} = [Fus3]_0 \frac{[\alpha]}{K_\alpha + [\alpha]}, \tag{4}$$

$$[Ste12]_{ss} = [Ste12]_0 \frac{[Fus3]}{K_3 + [Fus3]}, \tag{5}$$

$$[GFP]_{ss} = \frac{[GFP]_0}{1 + \frac{\Gamma[Fus3]_{ss}}{\gamma_{GFP}}} \frac{[Ste12]_{ss}}{K_{12} + [Ste12]_{ss}}. \tag{6}$$

We can combine these equations to obtain the steady state value of GFP as a function of $\alpha$ factor:

$$[GFP]_{ss} = [GFP]_0 \underbrace{\frac{[Ste12]_0[Fus3]_0}{K_3K_{12}} \frac{[\alpha]}{K_\alpha + [\alpha]\left(1 + \frac{[Fus3]_0}{K_3} + \frac{[Ste12]_0[Fus3]_0}{K_3K_{12}}\right)}}_{F(\alpha)} \underbrace{\frac{K_\alpha + [\alpha]}{K_\alpha + [\alpha]\left(1 + \frac{\Gamma[Fus3]_0}{\gamma_{GFP}}\right)}}_{G(\alpha)}. \tag{7}$$

$F(\alpha)$ is the steady state level of GFP that would be observed in a linear cascade without the repressive link from Fus3 to GFP (i.e. the system of **Figure 7A**). This function monotonously increases with increasing $\alpha$-factor concentration. $G(\alpha)$ captures the impact of degradation of Fus3 due to GFP.

The steady state level of GFP is thus of the general form

$$[GFP]_{ss} = B\frac{[\alpha]}{1 + C[\alpha]}\frac{1 + E[\alpha]}{1 + D[\alpha]}, \tag{8}$$

where the values of the constants $B$-$E$ can be derived from **Equation 7**. This function tends to zero for vanishing $[\alpha]$ and asympotically reaches the value $BE/(CD)$ for $[\alpha] \gg 1$. We note that the saturation at a non-zero value is consistent with our experimental observations (**Figure 7B**). Moreover, by taking the derivative it can also be shown that this expression has a maximum at an intermediate value of $[\alpha]$, consistent with the observed band-pass behavior.

## Data fitting

To fit our data we used a slightly more general expression where we allow for leaky production of GFP (paramter $A$) and also allow for a non-linearity in the cascade (Hill coefficient $n > 1$):

$$[GFP]_{ss} = A + B\frac{[\alpha]^n}{1 + C[\alpha]^n}\frac{1 + E[\alpha]}{1 + D[\alpha]}. \tag{9}$$

Parameters $A$, $B$, $C$, and $n$ are all shared between the data shown in **Figure 7A** (linear cascade) and **Figure 7B**. Parameter values used for the fits are $A$ = 1.1, $B$ = 8.6, $C$ = 0.8, $D$ = 9.9, $E$ = 0.2 and $n$ = 2.5.

