## [Decision Letter]

Thank you for submitting your article "Rewiring MAP kinases in *Saccharomyces cerevisiae* to regulate novel targets through ubiquitination" for consideration by *eLife*. Your article has been reviewed by two peer reviewers, and the evaluation has been overseen by Ivan Dikic as the Senior and Reviewing Editor.

The reviewers have discussed the reviews with one another and the Reviewing Editor has drafted this decision to help you prepare a revised submission.

Summary:

The ability to rationally engineer post-translational circuits utilizing processes like phosphorylation and ubiquitylation is critical for expanding synthetic biology beyond transcriptional circuitry. Here, the authors developed a new technique based on target phosphorylation motifs and protein interaction domains to engineer post-translational interactions between MAPK kinases and protein substrates. Using engineered pathways in yeast where a MAP kinase (Fus3 or Erk2) is targeted to artificial substrate protein through fusion of both the kinase and substrate to compatible interaction domains, they determine that this mode of interaction and the presence of the cognate phosphorylation motif (in this case a phosphodegron) is sufficient to confer MAPK regulation of the substrate (in this case leading to degradation). They used this technique to control ubiquitin-dependent degradation of fluorescent reporters (such as YFP) and to rewire signaling circuits in yeast both transcriptionally and post-translationally. The key findings include: 1) substrate phosphorylation by a MAPK kinase can be engineered in a highly modular fashion by combining two distinct modules (protein-protein interaction and phosphorylation motif); 2) the engineered phosphorylation can enable new signaling capabilities through feedback/feedforward loops. The technique is novel and is useful for dissecting natural signaling pathways as well as for synthesizing new signaling behaviors. In particular, it represents an exciting example of engineering a fully post-translational feedback loop in a signaling system, which makes this a major step forward from a synthetic biology point of view. The manuscript could be strengthened by consideration (and integration) of known mechanisms for docking interactions involving Fus3, and their role in regulating signal output (Remenyi et al. Mol Cell 2005).

Essential revisions:

1) Protein-protein interaction domains. The paper focused on exploring both natural and synthetic interaction domains, including PDZ, SH3, SYNZIP. In their approach, two interaction domains are fused separately to the MAPK kinase and the non-native protein substrate. Thus, both proteins need to be engineered to enable the phosphorylation interaction. In contrast to this approach (and as the authors citied), an alternative approach is to only engineer the protein substrate by utilizing native docking interactions between MAPK kinase and its substrate. Regot et al. and Durandau et al. used this alternative strategy to control fluorescent reporter localization. Can the authors comment on the relative capabilities of these different design strategies?

2) Related to point 1. In the current design, the ligand is fused to the kinase and the corresponding binding domain is fused to the substrate. What if the locations of the two domains are swapped? If the response is not sensitive to where the domain is fused, then it will provide stronger support to the claim that the design is highly modular. If the authors have data on this already, it would be helpful to report it because, even if the result is negative, it would be useful for other researchers seeking to design similar systems.

3) Additional controls, some biochemical in nature, should be provided to confirm that regulation is via the proposed mechanism. Kinase dead Fus3 should be included in certain key experiments (e.g. Figure 1) to confirm that phosphorylation of the substrate is actually mediated by the Fus3 kinase fusion. The authors should also confirm that degradation of the YFP-degron construct is mediated via the SCF complex and ubiquitin-mediated proteasomal degradation. The experiments describing implementation of negative feedback and feedforward topologies require the inclusion of dynamic, time-course data. Temporal regulation of signal output could be markedly different when MAPK is targeted to phosphorylate alternative network components.

4) Scalability of the design. In Figure 2, the authors used different interaction domains to control the phosphorylation of two different substrates by Fus3. However, the response of the YFP substrate is much weaker than that of the mCherry substrate. First, what about the response in individual cells? Is the response of YFP always weaker than mCherry in all individual cells, or only in subpopulations? Second, is this due to the saturation of the system? If so, it would be helpful to identify the source of limitation that can be improved in the future.

---

## [Author Response]

*Essential revisions:*

*1) Protein-protein interaction domains. The paper focused on exploring both natural and synthetic interaction domains, including PDZ, SH3, SYNZIP. In their approach, two interaction domains are fused separately to the MAPK kinase and the non-native protein substrate. Thus, both proteins need to be engineered to enable the phosphorylation interaction. In contrast to this approach (and as the authors citied), an alternative approach is to only engineer the protein substrate by utilizing native docking interactions between MAPK kinase and its substrate. Regot et al. and Durandau et al. used this alternative strategy to control fluorescent reporter localization. Can the authors comment on the relative capabilities of these different design strategies?*

We agree that this is an important point to emphasize, and we have more specifically contrasted the advantages of our strategy and those of Regot et al. and Durandau et al. in the introduction with the addition of this paragraph:

“Targeting a kinase to a new substrate is an essential step towards creating modular kinase signaling systems. As discussed above, Regot et al. and Durandau et al. have described an approach wherein a kinase-specific docking domain can be used to direct a particular kinase to a new substrate—a powerful tool for interrogating natural kinase signaling systems. However, the number of naturally occurring kinase-substrate docking interactions inherently limits the scalability of the approach. For example, a given kinase ‘module’ cannot be reused in parallel signaling pathways, because it would not be able to distinguish between downstream targets in 2 one pathway versus another. To overcome this limitation, it would be useful to be able to tease apart the ‘targeting’ module of the kinase from the ‘enzymatic’ module—and likewise, the ‘targeting’ and ‘effector’ modules of the substrate. If these functions can be defined as separable parts, the enzymatic module of a kinase would be available for reuse in orthogonal pathways, just by pairing it with unique targeting domains.”

We have also added the following sentence to the final paragraph of the discussion, to further highlight the benefits and limitations of both approaches:

“Thus, the scalability afforded by the use of modular interaction domains comes at the potential price of altered protein expression. In contrast, using docking domains for co-localization obviates engineering the kinase, but is not a scalable rewiring approach.”

*2) Related to point 1. In the current design, the ligand is fused to the kinase and the corresponding binding domain is fused to the substrate. What if the locations of the two domains are swapped? If the response is not sensitive to where the domain is fused, then it will provide stronger support to the claim that the design is highly modular. If the authors have data on this already, it would be helpful to report it because, even if the result is negative, it would be useful for other researchers seeking to design similar systems.*

On the advice of the reviewers we constructed and tested a system where the interaction domains fused to the kinase and substrate were exchanged. Specifically, we swapped the mPDZ domain from the substrate to the kinase and the PDZ ligand from the kinase to the substrate – the results can be found in Figure 1—figure supplement 3. The two orientations exhibited the same qualitative behaviors, however the maximal fold change observed for the swapped domains was approximately half that of the original orientation. These qualitative similarities – and quantitative differences – are consistent with our experiences with the other targeting domain pairs (Figure 2). As we have discussed in the manuscript, these differences are likely due the fusions affecting either protein expression or sterically interfering with the function of one of the involved enzymes. Thus, while these domains and their functions appear to be largely modular, it is not entirely cut and dry.

*3) Additional controls, some biochemical in nature, should be provided to confirm that regulation is via the proposed mechanism. Kinase dead Fus3 should be included in certain key experiments (e.g. Figure 1) to confirm that phosphorylation of the substrate is actually mediated by the Fus3 kinase fusion. The authors should also confirm that degradation of the YFP-degron construct is mediated via the SCF complex and ubiquitin-mediated proteasomal degradation.*

We agree that testing a kinase dead version of Fus3 is a thoughtful control, and that it would help demonstrate whether Fus3 kinase activity is necessary. We have performed the experiments described in Figure 1 again, this time with the addition of a strain expressing an inactive version of Fus3.

To provide the further biochemical evidence that the decrease of the YFP signal is mediated via proteasomal degradation, we have tested the protein level of the YFP-degron construct in the absence and the presence of the proteosomal inhibitor MG132 (Figure 1—figure supplement 2). The level of the YFP-degron fusion protein only decreases in cells treated with α-factor, not in cells treated 3 with both α-factor and MG132, indicating that normal proteasome function is critical for the turnover of the YFP signal.

*The experiments describing implementation of negative feedback and feedforward topologies require the inclusion of dynamic, time-course data. Temporal regulation of signal output could be markedly different when MAPK is targeted to phosphorylate alternative network components.*

On the reviewers’ advice, we have included time course data for both the negative feedback and the feed-forward topologies in Figure 6—figure supplement 1.

*4) Scalability of the design. In Figure 2, the authors used different interaction domains to control the phosphorylation of two different substrates by Fus3. However, the response of the YFP substrate is much weaker than that of the mCherry substrate. First, what about the response in individual cells? Is the response of YFP always weaker than mCherry in all individual cells, or only in subpopulations? Second, is this due to the saturation of the system? If so, it would be helpful to identify the source of limitation that can be improved in the future.*

We appreciate the reviewers’ concerns regarding the possibility that subpopulations of cells might be responding to the input signal in very different fashions. In fact, this is actually a concern throughout all of the experiments. For this reason we have recast all of our data to more clearly communicate the response across the entire population – instead of reporting the mean and standard deviations between separate experiments, the data now appear either as histograms or as the medians and quartiles of the populations. In all cases, the entire population appears to respond to the input – as opposed to a subset, which would result in a bimodal distribution, etc.

As suggested by the reviewers, we have performed experiments to determine whether a particular step in the system was limiting, and as a result caused the disparity between the degradation of mCherry and YFP shown in Figure 2. There are two points where components of our system might compete for limiting resources. Since the two copies of Fus3 are overexpressed, they may compete for access to the endogenously expressed upstream MAPKK, Ste7. Likewise, the YFP and mCherry substrates might compete for access to the ubiquitin/proteasomal machinery. To test this we constructed strains that expressed our standard system – one kinase targeting one substrate – and added either a competing copy of Fus3 or a competing substrate (Figure 3—figure supplement 1 and Figure 3—figure supplement 2). In both experiments we observed a diminished response in YFP degradation in the presence of the competitor. It should also be noted that the ABA-inducible interaction domains elicit a weaker degradation response compared to the mPDZ-PDZ ligand interaction (compare data in Figure 2). Thus, it is likely that a confluence of all these factors – saturation points as well as the less efficient ABA-induced interaction – contribute to the different levels of degradation observed for the mCherry and YFP reporters in this dual-targeting system.